**Data Availability Statement:** All accuracy raw data files are available from the DataVerse database (https://doi.org/10.5683/SP2/NTRNB7).

# Development and *ex-vivo* assessment of a novel patient specific guide and instrumentation system for minimally invasive total shoulder arthroplasty

Joshua W. Giles[1,2]*, Cyrus Brodén[3], Christine Tempelaere[3], Roger J. H. Emery[3], Ferdinando Rodriguez y Baena[2]*

**1** Department of Mechanical Engineering, University of Victoria, Victoria, British Columbia, Canada, **2** Department of Mechanical Engineering, Imperial College London, London, United Kingdom, **3** Department of Surgery and Cancer, Imperial College London, London, United Kingdom

\* jwgiles@uvic.ca (JWG); f.rodriguez@imperial.ac.uk (FRB)

## Abstract

### Objective

To develop and assess a novel guidance technique and instrumentation system for minimally invasive short-stemmed total shoulder arthroplasty that will help to reduce the complications associated with traditional open replacement such as poor muscle healing and neurovascular injury. We have answered key questions about the developed system including (1) can novel patient-specific guides be accurately registered and used within a minimally invasive environment?; (2) can accuracy similar to traditional techniques be achieved?

### Methods

A novel intra-articular patient-specific guide was developed for use with a new minimally invasive posterior surgical approach that guides bone preparation without requiring muscle resection or joint dislocation. Additionally, a novel set of instruments were developed to enable bone preparation within the minimally invasive environment. The full procedure was evaluated in six cadaveric shoulders, using digitizations to assess accuracy of each step.

### Results

Patient-specific guide registration accuracy in 3D translation was 2.2±1.2mm (RMSE±1 SD; p = 0.007) for the humeral component and 2.7±0.7mm (p<0.001) for the scapula component. Final implantation accuracy was 2.9±3.0mm (p = 0.066) in translation and 5.7–6.8 ±2.2–4.0˚ (0.001<p<0.009) across the humerus implants' three rotations. Similarly, the glenoid component's implantation accuracy was 3.0±1.7mm (p = 0.008) in translation and 2.3–4.3±2.2–4.4˚ (0.008<p<0.09) in rotation.

### Conclusion

This system achieves minimally invasive shoulder replacement with accuracy similar to traditional open techniques while avoiding common causes of complications.

**Funding:** Author FRyB received a Career Award from the Leverhulme Trust (https://www.leverhulme.ac.uk/). Authors FRyB and RJHE received funding through a Wellcome Trust Translation Fund award (Grant #098269/Z/12/Z; https://wellcome.org/grant-funding/schemes/translation-fund). The funders had no role in study design, data collection and analysis, decision to publish, or preparation of the manuscript.

**Competing interests:** 1. The authors JWG & FRyB have filed a patent in relation to the patient specific guide described in this study. Patent application number: 16710296.1-1122, and title: Patient-Specific Surgical Guide. This does not alter our adherence to PLOS ONE policies on sharing data and materials. 2. The authors RJHE & FRyB received funding from the Wellcome Trust for this work as part of a larger program of research. Funding number: 098269/Z/12/Z, and title: Total Shoulder Replacement System. The funder had no role in study design, data collection and analysis, decision to publish, or preparation of the manuscript. 3. Author FRyB received a Career Award from the Leverhulme Trust (https://www.leverhulme.ac.uk/). The funder had no role in study design, data collection and analysis, decision to publish, or preparation of the manuscript. 4. The authors FMRyB, RJHE, CB, & JWG hold an interest in a start-up company related to this work.

## Significance

This novel technique could lead to a paradigm shift in shoulder arthroplasty for patients with moderate arthritis, which could significantly improve rehabilitation and functional outcomes.

## Introduction

Total Shoulder Arthroplasty (TSA) has been shown to improve the function and minimize the pain of patients with joint degeneration [1–5]; however, up to 68% of patients exhibit subscapular dysfunction post-operatively due to subscapularis tenotomy or peeling performed during the deltopectoral approach, which can result in poor functional outcomes [6–11]. Additionally, the need for clear *en-face* access to the joint surfaces requires humeral dislocation, which has been linked to intraoperative nerve injury [12–14]. Furthermore, traditional surgical techniques have been shown to be prone to outliers in implantation accuracy that are associated with poor TSA outcomes [15–20]. These complications are particularly problematic in younger patients who have greater functional expectations and who necessitate longer implant survival [21, 22].

Attempts have been made to address these functional and survivorship shortcomings through the development of less invasive approaches and computer assisted technologies, respectively. Approaches that reduce invasiveness include splitting the subscapularis [23], limiting transection of the inferior subscapularis [24], splitting the deltoid and developing the rotator interval [25, 26], and approaching anterosuperiorly [27]. These methods have achieved variable reductions in invasiveness [28]; however, these techniques require humeral head dislocation and thus have the potential to cause neurologic complications [13]. Additionally, these approaches are more technically challenging with limited visualization, which, in at least one report, resulted in poor implant placement accuracy and sizing [26]. Computer assisted technologies, including patient specific guides (PSGs), have been developed to improve TSA accuracy [20, 29–33], but have only been used with invasive, open approaches. Thus, the current state-of-the-art in TSA can be characterized by a dichotomy between a) surgical techniques that reduce invasiveness in an attempt to improve function but potentially limit accuracy and impact implant survivorship, and b) computer assisted technologies that improve accuracy in an attempt to increase survivorship but use open techniques that are associated with functional and neurological complications.

To overcome these challenges and incorporate the advantages of both methods, we developed a novel guidance and instrumentation system (for short-stemmed TSA) that enables the use of a Minimally Invasive (MI), non-dislocating surgical approach that minimizes muscle disruption while facilitating accurate implantation. This technique is specifically focused on treating patients with moderate rather than end-stage arthritis as the former is a rapidly growing surgical population and moderate arthritis can be treated without extensive soft-tissue releases and resections, which lends itself to an MI approach [21, 22]. The goal of this *ex-vivo* study was to address four critical questions. (1) Can PSGs be accurately registered and effectively used within an MI surgical environment? (2) Can accuracy similar to open techniques be achieved when implanting a short-stemmed TSA within a MI environment? (3) What is the proximity between a trans-humeral drilling trajectory (see details in Methods) and the axillary nerve? (4) What surgical steps primarily contribute to final implantation error in this MI-TSA procedure?

## Methods

Below we describe the development of a PSG system and bone preparation instrumentation that enable the use of an MI, muscle-sparing, non-dislocated surgical approach [34] for short-

stemmed TSA. Subsequently, we describe the experimental testing protocol and statistics used to assess this technology.

## Surgical approach & procedural paradigm

Amirthanayagam et al's MI approach [34], developed in an earlier stage of our research program, consists of a posterior skin incision, originating just below the acromial angle, that extends ~10 cm inferiorly (Fig 1). By cutting between and retracting the infraspinatus and teres minor muscle groups, this approach avoids the muscular transection that occurs with other approaches. This in turn reduces recovery time and prevents muscular insufficiency [6–8], thus enhancing post-operative function. Furthermore, the approach avoids glenohumeral dislocation, which has been shown to cause neurovascular injury that decreases post-operative function [14]. However, not dislocating the joint when using this MI approach prevents the traditionally required *en-face* articular access; thus, a new paradigm and instrumentation are required for bone preparation and implantation.

The proposed paradigm involves simultaneously creating a trans-humeral tunnel and glenoid guide hole (see S1 Animation for a computer rendered video animation of the process). The tunnel guides humeral preparation (e.g. cutting/reaming) and acts as a working channel for power drivers to be passed into the joint, thus avoiding the need for *en-face* articular access. A similar humeral tunnel concept is used for humeral head focal defect arthroplasty in the Partial Eclipse system (Arthrex, Naples, USA) [35]; however, that system is unable to accurately create a glenoid guide hole, and thus cannot be used for TSA.

## Novel PSG & instruments

The following is an overview of the PSG and instruments required for this MI technique. For full design and functional details, please see the S1 Appendix, as well as a demonstration of the process to use this technique in the S1 Animation).

To achieve the above procedural paradigm, we developed a PSG (Fig 2A–2D) (Patent Application: 16710296.1–1122) to guide simultaneous drilling of the trans-humeral tunnel and the glenoid guide hole. Drilling from the lateral humerus and into the scapula is achieved

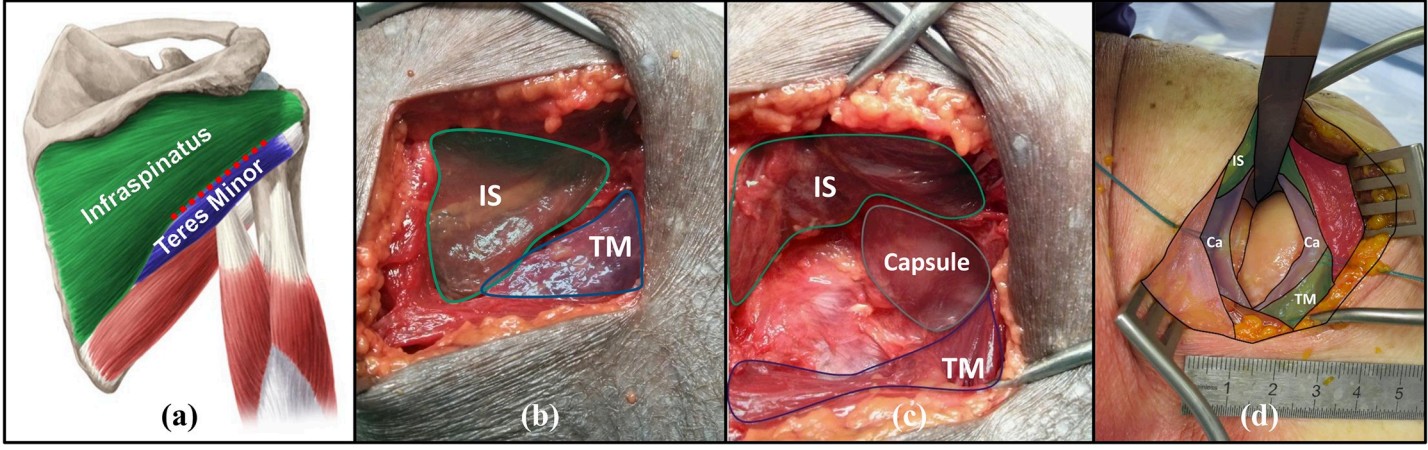

**Fig 1. Diagram and photos illustrating the posterior muscle splitting approach.** (a) Posterior view of the shoulder with the muscle splitting incision (dashed red line) indicated between the infraspinatus (IS) (green) and teres minor (TM) (blue). (b) Photograph following initial skin incision with shaded regions indicating IS & TM muscle bellies. (c) Photograph of split and retracted IS & TM muscle bellies with joint capsule (Ca) also shaded. (d) Photograph showing incised capsule with articular surfaces visible within the joint.

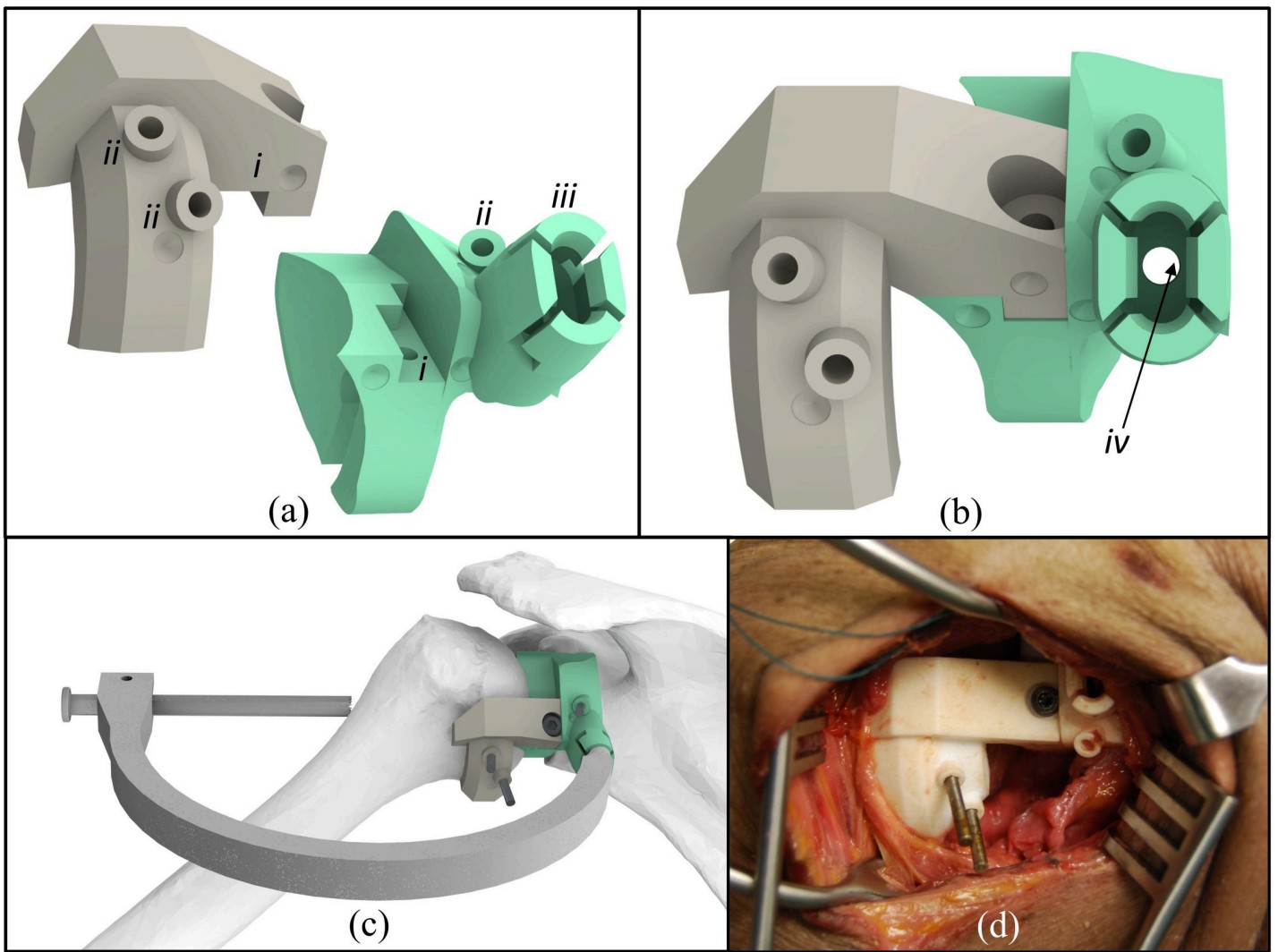

**Fig 2. Computer renderings and photo of two-sided minimally invasive PSG.** (a) Two PSG components from posterior-inferior view before being connected together with critical features including inter-lock mechanism (i), fixation pin holes (ii), and a standardized snap-fit feature for drill guide attachment (iii). (b) Posterior view with components connected showing hole used to guide creation of reference mark on glenoid rim (iv). (c) PSG registered to the humerus and scapula, such that they are placed in a pre-operatively planned pose with attached c-shaped drill guide that enables simultaneous trans-humeral and glenoid guide hole creation, with fixation pins inserted through guides into underlying bone, and a bolt rigidly fixing the patient specific guide components together, (d) photo showing PSGs registered to bones within posterior muscle splitting surgical approach.

using a custom c-shaped drill guide (Fig 2C) that attaches to the PSG by way of a snap-fit feature (Fig 2A(iii)) that is pre-operatively designed into the guide. The PSG is composed of two sub-components that are connected together after each is rigidly fixed to its respective bone (Fig 2A). These components are designed pre-operatively using a patient CT scan such that when they are properly attached to each bone they coaxially align their central guide axes; these axes are commonly used in TSA as references for implant positioning and orientation. By incorporating anatomical features of both bones into one PSG, the guide can lock the two bones together in this pre-operatively defined configuration. Additionally, using a muscle-sparing surgical approach enhances PSG physical registration (i.e. locking) to the bones due to joint compression caused by passive tension in the intact rotator cuff. Please see the S1 Appendix for details of pre-operative PSG design.

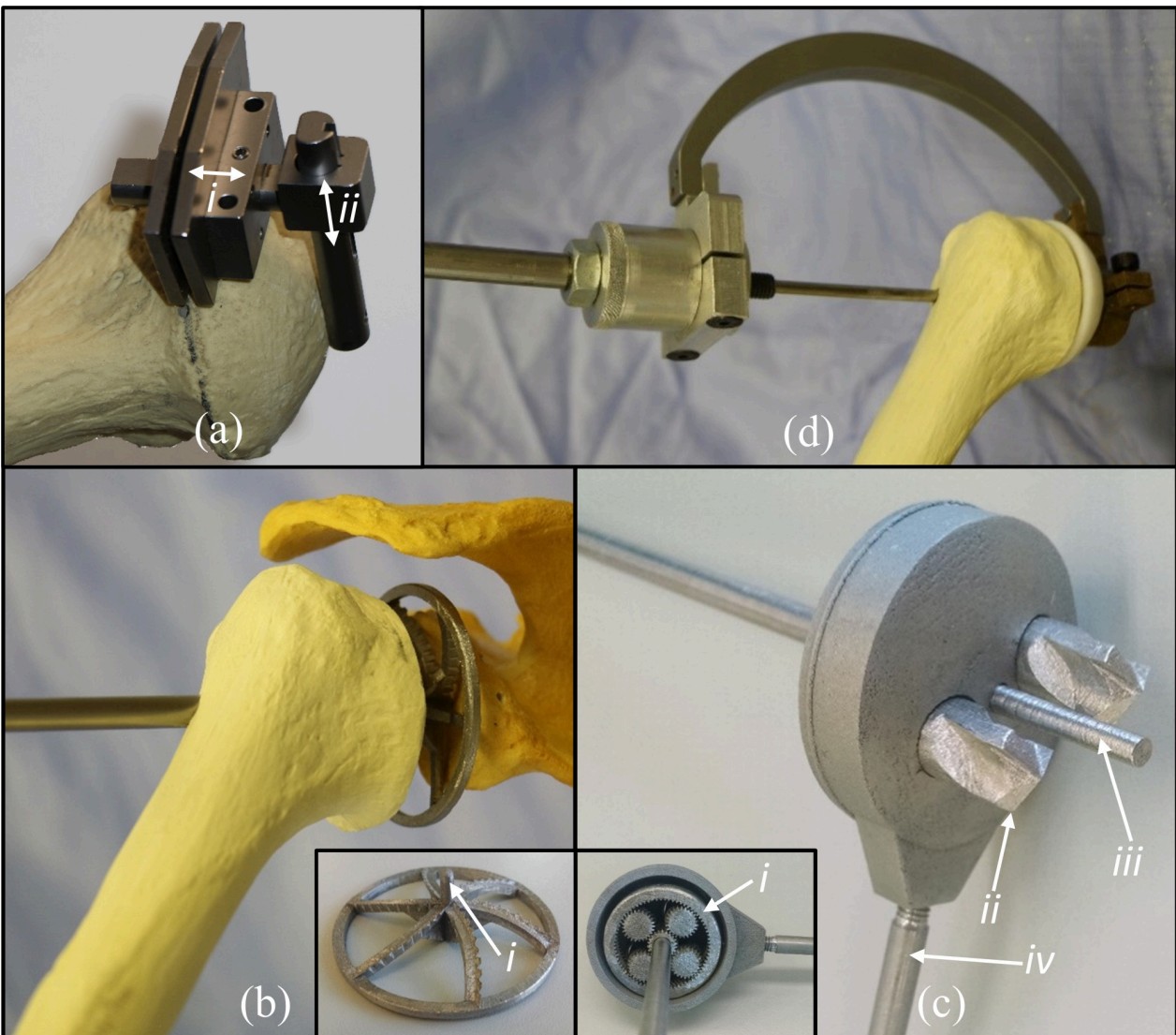

**Fig 3. Computer renderings and photos of the minimally invasive instrumentation.** (a) Humeral head resection guide with head height (i) and diameter (ii) adjustability. (b) Foam bone demonstration of glenoid reamer within the joint connected to trans-humerally inserted driver with an inset image of the reamer showing the guide pin (i). (c) Drill bit (ii) mechanism for creation of glenoid peg with inset image of epicyclic gear system (i), guide pin (iii), and handle (iv). (d) Slap hammer impactor configured to impact humeral trial, attached to humerus with guide pin inserted into guide hole, and shaft attached (note: slap mass out of view).

### Minimally invasive instrumentation

**Humeral cutting guide.**   The humeral implant used in this study (Affinis, Mathys Ltd, Bettlach, CH) requires humeral head resection; however, estimating retroversion by visualizing the articular margins is not achievable within an MI environment. Therefore, we designed a cutting guide (Fig 3A) that sets the cut plane orientation by referencing the previously drilled humeral guide tunnel, which, by design, is perpendicular to the humeral resection plane. The cutting guide can be adjusted to work with a wide range of humeral head heights/diameters (Fig 3A(i) and 3A(ii)).

**Glenoid reamer.**   As there is not room for a glenoid-fixed guide pin within the MI environment, the standard cannulated reamer was replaced with a custom design that incorporates

a guide pin on its cutting face (Fig 3B(i)), which is inserted into the glenoid guide hole and powered by a driver inserted through the humeral tunnel.

**Off-axis drilling mechanism.** The glenoid peg holes necessary for glenoid implant impaction cannot be created within an MI environment using traditional drilling that requires unobstructed perpendicular glenoid access. To overcome this, a novel surgical drilling tool was designed that is powered by a centrally located driver passed through the humerus but which can drill holes that are radially displaced from the driver's axis. This tool is composed of a stainless steel additively manufactured epicyclic gear mechanism with drill bits integrated into two of the planet gears (Fig 3C(i) and 3C(ii)). A housing fits around the gear system (with the drill bits protruding through its base–Fig 3C(iii)) and constrains the planet gears such that each rotates about its axis without precessing; thus, cylindrical holes are produced. The powered driver is designed to extend beyond the drill bit tips and is inserted into the glenoid guide hole thus controlling the position, version, and inclination of the drill holes. The drill hole rotation around the driver-axis is controlled by a handle (Fig 3C(iv)) that the surgeon aligns with a reference mark on the posterior glenoid vault that is created while the PSG is fixed to the scapula (Fig 2B(iv)).

**Implant impactors.** To enable glenoid implant impaction, a c-shaped impactor was designed that allows a contoured impaction head to be inserted through the posterior incision while avoiding the undislocated humeral head. The humeral stem and ceramic head can be impacted directly by a surgical mallet or by using a custom slap hammer impactor if the joint has insufficient internal rotation (please see the S1 Appendix for full details). The impactor is used to sequentially impact the Affinis implant cruciform-shaped cutters, compactors, and final stem, as well as humeral head trials and the final ceramic component.

## Testing protocol

Six mildly osteoarthritic shoulder specimens (unpaired, 3 left, age: 72.2±4.3 years) with intact rotator cuffs, and full scapulae and humeri were tested using our MI technique and instruments (Ethical Approval # 13/LO/1839). CT scans we obtained and used to create the PSGs for each specimen. No control specimens, treated using a traditional open surgical technique, were tested in this study as significant data are available in the literature [20, 29–33, 36, 37] and thus the use of additional cadaveric remains could not be ethically justified. The distal third of the humerus and medial border of the scapula were denuded to allow the attachment of tracker pins, collection of surface geometry, and clamping of the bones during testing. Infrared trackers (Optotrak Certus, NDI, Waterloo, CA) were pinned to the distal humerus and scapular spine. The MI, posterior surgical approach was then conducted as described above. After accessing the joint space, soft tissues including the labrum and cartilage located on the humerus and scapula within the PSG footprint were removed. The glenoid fossa, posterior vault, medial scapular spine, and subscapular, infraspinatus, and supraspinatus fossae were digitized with respect to the scapula tracker. The posterior humeral head/neck, distal humeral articular surfaces, and metaphysis were digitized with respect to the humeral tracker. These digitizations were only used for experimental purposes to enable comparison to the pre-operative plan as explained below.

The MI-PSG humeral component was physically registered (i.e. articulated with the bony anatomy matching the PSG's geometry) and pinned in place. The PSG main body was inserted between the glenohumeral articular surfaces with its posterior aspect seated on the glenoid vault. If, with the PSG in place, the rotator cuff exhibited laxity, it was removed and replaced by a 5 mm thicker version of the PSG; this process is akin to using a thicker tibial implant to remove laxity in knee replacement. Once seated, the PSG was pinned in place. The arm was then oriented in ~60° abduction, neutral axial rotation, and extension (i.e. the planned arm

orientation) and the humeral head was manipulated until the two PSG parts interlocked and could be fixed together. A surgical limb positioner was then attached to the arm to lock it during the following steps.

A 2.5 mm drill bit–directed by the PSG guide hole at the base of the snap-fit mating feature (Fig 2B(iv))–was used to create a glenoid vault reference mark to be used during glenoid peg hole drilling †. The c-shaped drill guide was then inserted into the snap-fit feature and the guide's cannula was slid to the arm's lateral skin surface. The cannula's contact point on the skin was used as the center for a 3 cm superoinferior incision. The incision was developed using blunt dissection until the lateral humeral cortex was reached and the axillary nerve was localized. During dissection, the shoulder was maintained in the planned orientation to ensure that the nerve's path remained as it would be during surgery. The cannula was then advanced to the bone surface and locked. A 3 mm drill bit was advanced through the humeral head, the PSG clearance hole, and into the glenoid to create the humeral and glenoid guide holes. Subsequently, a 4 mm drill bit was used to widen the humeral guide hole to enable sufficiently large power drivers to be used during reaming and drilling. The cannula was then withdrawn, the PSG fixation pins extracted, and the PSG removed.

The humeral cutting guide was placed into the joint and its guide pin inserted into the guide tunnel. The cutting block was adjusted to the humeral head size (adjustment described in S1 Appendix), locked, and two diverging pins inserted. The head was resected using an oscillating saw. After removing the cutting guide and resected head, an appropriately sized glenoid reamer was inserted and attached to a trans-humeral driver. The reamer's guide pin was inserted into the glenoid guide hole and the glenoid was reamed just until the subchondral bone was reached. Next, the epicyclic drilling system was placed into the joint and a trans-humeral driver was passed through it and inserted into the glenoid guide hole. To control the drill system's orientation around the driver axis, the system's handle was aligned with the previously created reference mark on the posterior glenoid vault (see † in previous column). The system was then driven and advanced until its housing contacted the reamed glenoid indicating the peg holes were fully drilled.

A 3D-printed replica of the Affinis glenoid implant was then impacted. A replica was used as it allowed precise fiducial features to be incorporated into its geometry, thus enabling the implant pose to be accurately measured. Finally, the humeral metaphysis was prepared by fixing the Mathys Humeral Positioning Disc (see the product manual [38]), and impacting the Pre-Impactor and Impactor tools, which create a cruciform shaped recess. After using a trial to size the humeral head component, the stem and humeral head were impacted.

## Outcome measures

To answer our four questions, digitizations were taken after completion of each of these steps: i) registration of the PSGs, ii) drilling of the humeral and scapular guide holes, iii) humeral head resection, iv) final glenoid implant placement, v) final humeral stem placement. As well, qualitative observations were made of the PSG usability within the MI environment.

**Axillary nerve proximity.** To assess the proximity of the humeral guide tunnel's entry point on the lateral humerus to the path of the axillary nerve (Question 3) the nerve's path was digitized after it was localized within the lateral incision. Trans-humeral drilling was then conducted and the resulting entry point on the lateral humerus was digitized. The minimum distance between the nerve path and the entry point was calculated. If the nerve was not visible within the surgical field, a value of 15 mm was assumed because this is the amount the ~30 mm incision extended superior and inferior from the entry hole and thus is a conservative estimate of the nerve's minimum proximity.

**Experimental to pre-operative plan registration.** To determine the experimentally collected data's accuracy compared to the pre-operative plan, the Iterative Closest Point (ICP) registration algorithm was used [39]. Specifically, the surface digitizations of the two bones recorded at the beginning of the protocol were registered to the 3D bone models in their pre-operatively planned pose (please see the S1 Appendix). The spatial transformation matrix describing this registration was used to transform all outcome measure digitizations to the pre-operative plan coordinate system for comparison.

**Accuracy of PSG registration & implant placement.** To assess the accuracy with which the two PSG components were physically registered to their respective bones and the accuracy with which the humeral and glenoid implants were placed, fiducial points were incorporated into each component (Fig 2). The fiducials were placed such that an optically-tracked stylus could digitize them during the procedure and the points would construct a component coordinate system. These coordinate systems were transformed using the established registration matrix and compared to the pre-operative plan. For each of the two PSG components and two implants, this yielded a 3D translational error and three rotational errors with respect to the pre-operative plan coordinate system.

**Accuracy of intermediate steps.** The accuracy of intermediate steps was also assessed by digitizing key landmarks, instrument features, and/or surgically created features (e.g. drill hole entrance/exit points), transforming these to the pre-operative plan, and computing comparative translational/rotational errors. Please see the S2 Appendix for details on the process for each step.

**Statistical & power analyses.** From a clinical feasibility perspective, it is most important to compare the resulting error values to error/accuracy values reported in the literature for previous methods as is done in the Discussion section. However, from a technical development perspective, it is also important to compare the error values to the pre-operatively planned target placement. Therefore, for the PSG and final implant accuracy outcomes, single sample t-tests were conducted to determine if a statistically significant difference existed between the measured error values and a value of zero error (i.e. the pre-operative plan). Note that a significant difference only indicates the result statistically differs from the pre-operative plan but not that the error is clinically unacceptable. An *a-priori* sample size calculation was conducted using pilot standard deviations of $\leq 1.6$ mm and $\leq 4.4°$, and determined that six specimens would achieve >80% power with clinically meaningful differences of 5 mm and 7.5°, which are lower bound estimates of errors that meaningfully impact biomechanics [40–42]. To understand each surgical step's contribution to final implantation error, paired t-tests were used to compare each step to its predecessor. Standard deviations of each surgical step were not recorded during pilot testing. Thus, to conduct a sample size calculation for the paired t-tests, an assumption was made that each intermediate step had half the standard deviation of the final implantation pilot data and that stepwise differences of 2.5mm and 3.75° (i.e. half the clinically meaningful differences for final implantation error) are meaningful. Under these assumptions, fewer than six specimens were needed to provide 80% power.

## Results

### Question 1: PSG accuracy

The main PSG component registered to the glenoid with an accuracy (mean±SD) of 2.7±0.7 mm in 3D translation (p<0.001) and 0.9±0.8° (p = 0.046), 3.1±1.8° (p = 0.009), and 4.5±2.8° (p = 0.012) in rotation about the scapula's medial-lateral, superior-inferior, and anterior-posterior axes, respectively. Additionally, the secondary PSG component registered to the humerus with an accuracy of 2.2±1.2 mm in 3D translation (p = 0.007) and 8.2±5.0° (p = 0.01), 4.9±2.7°

(p = 0.007), and 6.0±6.0˚ (p = 0.056) in rotation about the humerus' medial-lateral, superior-inferior, and anterior-posterior axes, respectively.

### Question 2: Final implantation accuracy

The accuracy of the final glenoid component positioning and rotations about the scapula's medial-lateral, superior-inferior, and anterior-posterior axes was found to be 3.0±1.7 mm (p = 0.008), 2.3±2.2˚ (p = 0.051), 4.3±4.4˚ (p = 0.090), 3.4±3.1˚ (p = 0.045), respectively (Fig 4). After impacting the humeral cruciform stem, the accuracy of its positioning, and inclination and version rotations were found to be 2.9±3.0 mm (p = 0.066), 6.8±4.0˚ (p = 0.009), and 5.7 ±2.2˚ (p = 0.001), respectively (Fig 4).

### Question 3: Nerve proximity

The minimum distance between the axillary nerve's path and the humeral entry hole was on average 9.2±5.8 mm with a range of 0 mm (i.e. nerve overlapping the intended hole location) to >15 mm (i.e. nerve outside field of incision).

### Question 4: Stepwise accuracies

The accuracy at intermediate steps after PSG registration are as follows. The drilled humerus and scapula guide holes had 3D rotational accuracies (i.e. combined version and inclination error) of 6.4±3.2˚ and 8.2±5.2˚, respectively, which did not differ from the 3D rotational errors of the PSGs (see Question 1 results; p = 0.403 & p = 0.319). The positional error of the humeral guide hole on the articular surface relative to the pre-operative plan was 2.9±1.2 mm, which did not differ from the error introduced by humeral PSG registration (p = 0.272). The accuracy of the guide hole on the glenoid articular surface was 2.8±1.3 mm, which did not differ from the error introduced by glenoid PSG registration (p = 0.813). After resecting the humeral head, the 3D angular accuracy of the cut plane was found to be 8.4±3.7˚, which did not differ from the humeral guide hole angular error (p = 0.351). The accuracy of the final humerus component implantation (see Question 2 results) did not differ from the positional errors after humeral guide hole drilling (p = 0.989) or the angular errors after humeral head resection (p = 0.229). The accuracy of the final glenoid component implantation (see data in Question 2 results), did not differ from the positional or rotational errors after the glenoid guide hole was created (p = 0.669 & p = 0.319).

## Discussion

To avoid the highly invasive steps of muscle resection and joint dislocation required in traditional open surgical techniques, this work developed a method and set of instruments that enable implantation of a short stemmed TSA through a muscle-sparing, non-dislocating MI approach. This study addressed important questions regarding this technique's efficacy, accuracy, and feasibility. The findings indicate that appropriately designed PSGs and instruments can be used to achieve accurate implantation within an MI environment.

### Limitations

This study has limitations. First, the specimens did not have the moderate arthritic changes this technique was targeted towards; instead, all specimens exhibited mild arthritic changes (e.g. intact articular cartilage with some osteophytes). Shoulders with greater arthritis are often stiffer, which may make the procedure more challenging; however, this technique and its instruments were designed for that condition. Conversely, the osteophytes in a more arthritic

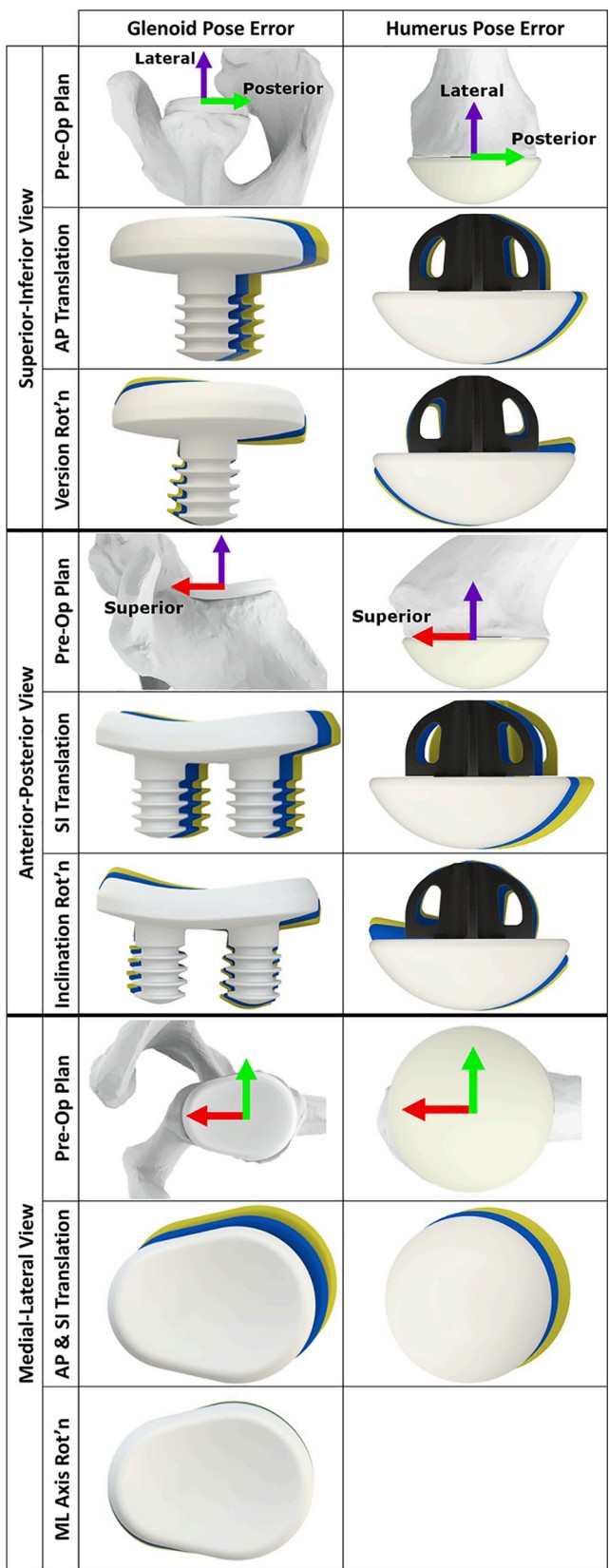

**Fig 4. Computer renderings of final glenoid and humeral implant placement error.** This figure is significant as it demonstrates the magnitude of the implantation errors relative to the respective bones. The first and second major columns show the translational and rotational errors of the glenoid and humerus implants, respectively. The 1st, 4th, and 7th major rows show the pre-operatively planned implant placements with the scapula and humerus bones shown for reference. Each image shows the pre-operatively planned position (white implant), the average absolute error (blue implant), and the upper bound of the 95% confidence interval (yellow implant). Note: the bones shown are approximately average for the specimen population. Rotational error around the medial-lateral axis are not shown for the axisymmetric humeral implant.

shoulder provide unique geometry, which would improve PSG registration accuracy as demonstrated by Van den Broeck [43]. Therefore, it is expected that the technique may prove more accurate when applied to the intended arthritic condition. Second, specimens did not include the forearm, which if included, may have negatively affected PSG fixation due to the additional weight; however, our use of a limb positioner likely mitigated this factor. Third, a control group of specimens treated with a traditional open surgical technique was not included in this study as significant data are available in the literature detailing the accuracy of these procedures and thus it was not thought appropriate to use additional cadaveric specimens to obtain similar data. However, this decision did limit the ability to statistically compare the developed method to traditional approaches and to collect data on the accuracy of each surgical step, which are not available in the literature. Despite this, the data available from the literature was sufficient to evaluate the overall accuracy of the developed technique.

## Question 1: PSG usability & accuracy

**Usability.** A number of important usability observations were made. First, increasing PSG thickness improved registration by increasing passive joint compression. Second, once the PSG components were connected, the humerus and scapula were observed to be rigidly linked together; however, it was necessary to use a limb positioner to prevent the weight of the arm shifting the PSGs during drilling. Third, in moderately arthritic specimens with osteophytes (used during pilot testing), the PSGs uniquely registered (i.e. 'locked') to the bones; however, the specimens in this study had mild arthritis and thus had a less obvious inter-lock. As a result, 3D-printed bone surrogates were used as an intra-operative visual reference to help verify proper PSG positioning on each specimen's bones.

**Accuracy.** Although most of the statistical tests of PSG registration accuracy were significant ($p < 0.05$), meaning they were different from the pre-operative plan, they were still deemed clinically acceptable for a number of reasons. First, the average translational (2.2–2.7 mm) and rotational (0.9–6.0˚) errors were all below the implantation error limits defined in the literature as yielding meaningful biomechanical effects except the humeral PSG rotation about the lateral axis (8.2˚), for which no threshold has been defined. Second, although no previous paper has directly measured PSG registration accuracy, studies have assessed the accuracy of drilling a central guide hole/pin using a PSG, which can be used as a surrogate for PSG registration accuracy. Levy et al. [44] found guide pin errors of up to 2.1 mm and 8.4˚, which are somewhat smaller than our own but were achieved using an open deltopectoral approach rather than an MI approach as was done here. Thus, the described PSGs produce similar accuracy to current clinical techniques but using an MI method.

## Question 2: Final implantation accuracy

The final implantation accuracy of this new MI technique must be placed in the context of previous reports. A recent meta-analysis by Villatte et al. reported the accuracy of glenoid component implantation through a deltopectoral approach using standard instrumentation or PSGs

to be: Standard Instruments—6.9˚ (95% CI: 2.7˚ to 11.1˚) for version, 11.6˚ (95% CI: 7.4˚ to 15.8˚) for inclination, 2.1 mm (95% CI: 1.6 mm to 2.6 mm) for positioning; PSG—2.9˚ (95% CI: 1.2˚ to 4.6˚) for version, 1.7˚ (95% CI: 0.5˚ to 3.0˚) for inclination, 1.5 mm (95% CI: 1.2 mm to 1.8 mm) for positioning [37]. In comparison, this MI technique yielded rotational accuracies of 4.3˚ (95% CI: 0.3˚ to 8.9˚) for version, 3.4˚ (95% CI: 0.14˚ to 8.9˚) for inclination, and a translational accuracy of 3.0 mm (95% CI: 1.2 mm to 4.8 mm). Thus, this MI technique achieves greater rotational accuracy and less variability compared to standard instrumentation with an open approach. With respect to translation, the MI technique compared to standard instrumentation has somewhat lower accuracy (3.0 mm vs 2.1 mm). However, the authors consider one specimen's translation an outlier, which if removed, results in an accuracy of 2.4 mm (95% CI: 1.1–3.7), which is similar to traditional instruments. This MI technique's results are however moderately poorer than Villatte's PSG-assisted accuracy for open techniques. No literature exists on humeral component implantation accuracy using PSGs and thus direct comparison is not possible; however, the measured accuracies are less than error values (5 mm and 7.5˚) that are thought to be biomechanically significant [40–42]. Thus, the developed technique achieves accuracy similar to accepted techniques while benefiting from the advantages of an MI approach.

## Question 3: Nerve proximity

The observed axillary-nerve-to-guide-hole distance was on average 9.2 mm. These results are in agreement with Prince et al. [45] who found an average distance of 7 mm from the nerve to the head of a proximally-directed intramedullary nail fixation screw that has a trajectory similar to our guide axis. Therefore, this technique may risk nerve injury and thus, use of a stab wound incision for trans-humeral drilling is not appropriate. Instead, as in this study, a ~3cm incision should be made followed by blunt dissection and placement of a cannula against the humeral cortex to ensure protection of the nerve.

## Question 4: Stepwise accuracy

Important insights can be gained by looking at which surgical steps primarily contribute to procedural error. First, the strongest influence on final implant positional accuracy is initial PSG registration as no intermediate step produced a statistically significant change in error (Glenoid: PSG—2.7 mm, Guide hole– 2.8 mm, Implant– 3.0 mm & Humerus: PSG—2.2 mm, Guide hole– 2.9 mm, Implant– 2.9 mm). Similarly, the 3D rotational error of the final implant orientation was most strongly influenced by the PSG registration error as no intermediate step produced a statistically significant change in error (Glenoid: PSG– 6.1˚, Guide hole– 8.2˚, Implant– 6.1˚ & Humerus: PSG– 8.3˚, Guide hole– 6.4˚, Head Resection– 8.4˚, Implant– 9.2˚). Two important points can be drawn from this: 1) as is clinically understood with existing PSGs for open surgery [33], it is critical that any soft tissues lying within the PSGs footprint be thoroughly removed because, if left in place, they will strongly affect PSG registration and thus implantation accuracy; 2) improvements in PSG registration accuracy in patients with more advanced arthritis due to their unique bone geometry (e.g. osteophytes) are likely to improve implantation accuracy. This latter point is supported by analysis of Hendel et al.'s data [33], where increasing glenoid deformity correlated with improved implant placement accuracy when using PSGs in open surgery. A second important finding is that although creation of the humerus and glenoid guide holes using trans-humeral drilling is not a significant source of error on average, it is prone to outliers. Specifically, for one specimen, there was a large increase in angular error between PSG registration (~5˚) and scapula guide hole drilling (~17˚), due to the glenoid PSG shifting before the limb positioner was attached. This

emphasizes the importance of supporting the arm's weight between PSG registration and limb positioner attachment. Removal of this outlier would result in an average error of only 5.3 ±3.7˚. Third, glenoid implant peg hole drilling using the drilling mechanism (Fig 3C) is the only surgical step that dictates glenoid implant error about the lateral axis and produced an accuracy of 2.5±3.0˚, which is well below the clinically meaningful stepwise error threshold defined in the power analysis. Thus, this novel mechanism, which enables simultaneous multi-peg hole drilling within the MI environment, is a viable tool for TSA.

## Conclusion

This PSG and instrumentation system can guide TSA through a minimally invasive approach with an accuracy equal to or better than traditional highly invasive, unassisted techniques. It avoids muscular transection and joint dislocation, which respectively risk poor healing and neurovascular injury, and as a result has the potential to reduce rehabilitation time and enhance functional outcomes. However, in using this technique, care must be taken to avoid axillary nerve injury on the lateral humerus. With these potential benefits and the current results in mind, this novel technique and technology has the potential to shift the paradigm in TSA for patients with moderate arthritis.

## Supporting information

**S1 Animation. Computer rendered animation of patient specific guided minimally invasive total shoulder arthroplasty.** This animation video shows the first stage of the developed minimally invasive total shoulder arthroplasty technique from patient specific guide insertion through trans-humeral drilling and guide removal.
(MP4)

**S1 Appendix. Detailed descriptions of novel instruments.** This appendix provides detailed descriptions of the various instruments used in the minimally invasive shoulder replacement procedure and in the case of the patient specific guides, how they are created.
(DOCX)

**S2 Appendix. Processes for assessing accuracy of intermediate steps.** This appendix provides details on how the accuracy of each surgical step in the procedure was measured and assessed.
(DOCX)

## Author Contributions

**Conceptualization:** Joshua W. Giles, Roger J. H. Emery, Ferdinando Rodriguez y Baena.

**Data curation:** Joshua W. Giles.

**Formal analysis:** Joshua W. Giles, Ferdinando Rodriguez y Baena.

**Funding acquisition:** Roger J. H. Emery, Ferdinando Rodriguez y Baena.

**Investigation:** Joshua W. Giles, Cyrus Brodén, Christine Tempelaere, Ferdinando Rodriguez y Baena.

**Methodology:** Joshua W. Giles, Roger J. H. Emery, Ferdinando Rodriguez y Baena.

**Validation:** Joshua W. Giles.

**Visualization:** Joshua W. Giles.

**Writing – original draft:** Joshua W. Giles, Cyrus Brodén, Christine Tempelaere, Roger J. H. Emery, Ferdinando Rodriguez y Baena.

**Writing – review & editing:** Joshua W. Giles, Ferdinando Rodriguez y Baena.

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
