## [Decision Letter · Decision Letter 0]

3 Mar 2021

PONE-D-20-37057

Development and ex-vivo assessment of a novel patient specific guide and instrumentation system for minimally invasive total shoulder arthroplasty

PLOS ONE

Dear Dr. Giles,

Thank you for submitting your manuscript to PLOS ONE. After careful consideration, we feel that it has merit but does not fully meet PLOS ONE’s publication criteria as it currently stands. Therefore, we invite you to submit a revised version of the manuscript that addresses the points raised during the review process.

The reviewer panel was enthusiastic about this manuscript, but some important concerns arose about the cohort and lack of a control group. Those should be addressed in your revision.

We look forward to receiving your revised manuscript.

Kind regards,

Alejandro A. Espinoza Orías, PhD

Academic Editor

PLOS ONE

Journal Requirements:

2)  We note that you have a patent relating to material pertinent to this article. Please provide an amended statement of Competing Interests to declare this patent (with details including name and number), along with any other relevant declarations relating to employment, consultancy, patents, products in development or modified products etc. Please confirm that this does not alter your adherence to all PLOS ONE policies on sharing data and materials, as detailed online in our guide for authors http://journals.plos.org/plosone/s/competing-interests by including the following statement: "This does not alter our adherence to  PLOS ONE policies on sharing data and materials.” If there are restrictions on sharing of data and/or materials, please state these. Please note that we cannot proceed with consideration of your article until this information has been declared.

3) We note that you have stated that you will provide repository information for your data at acceptance. Should your manuscript be accepted for publication, we will hold it until you provide the relevant accession numbers or DOIs necessary to access your data. If you wish to make changes to your Data Availability statement, please describe these changes in your cover letter and we will update your Data Availability statement to reflect the information you provide.

4) Please include captions for your Supporting Information files at the end of your manuscript, and update any in-text citations to match accordingly. Please see our Supporting Information guidelines for more information: http://journals.plos.org/plosone/s/supporting-information.

Reviewers' comments:

Reviewer's Responses to Questions

**Comments to the Author**

1. Is the manuscript technically sound, and do the data support the conclusions?

Reviewer #1: Partly

Reviewer #2: Yes

2. Has the statistical analysis been performed appropriately and rigorously? 

Reviewer #1: Yes

Reviewer #2: Yes

3. Have the authors made all data underlying the findings in their manuscript fully available?

Reviewer #1: Yes

Reviewer #2: Yes

4. Is the manuscript presented in an intelligible fashion and written in standard English?

Reviewer #1: Yes

Reviewer #2: Yes

5. Review Comments to the Author

Reviewer #1: Thank you for the opportunity to review this paper.

1. Introduction: Clear but could be shortened to be more concise.

2. Methods: Clear. Power analysis performed. It is this reviewer's opinion that the one sample t-tests adds limited value. Instead, as noted by the authors, is it possible to rather evaluate accuracy in terms of clinical relevance?

3. Results: Clear and concise. See comment with regards to the p-values in the methods section.

4. Discussion: A major limitation of the study is the lack of a control group for comparison of the achieved accuracy between the minimal invasive and invasive approaches. This should be highlighted in the text. The discussion is somewhat long and can be shortened without loosing any relevance.

Reviewer #2: Dear author,

thank you for your interesting paper about an innovative surgical technic of shoulder prosthesis implantation. However the application field in terms of indications seems limited (Young patients with mild OA) and not sure that the accuracy provided by the system is so acceptable than you concluded.

I don't have any comments for the methodology that is quite rigorous, except for humeral rotation assessment that is unaccurate... or for other details...

I have few remarks about the form:

- L30-31: SD values are missing

- L33: change "equal" for similar

- L81-86: these sentences could be removed because of the redundancy with the introduction especially

- Fig 2: please flip the picture b, and change the arrangement of the 2 pieces of the PSG on the picture a, for easing the understanding of the operation of the PSG fixation (to get the same orientation of the visualization)

- Fig 3: same comment for the picture b

The nerve proximity at the level of the glenoid neck would have been interesting to assess too because the fixation of the PSG looks very prominent in this area...

6. PLOS authors have the option to publish the peer review history of their article (what does this mean?). If published, this will include your full peer review and any attached files.

Reviewer #1: No

Reviewer #2: No

---

## [Author Response · Author response to Decision Letter 0]

12 Apr 2021

Responses to Reviewer Comments:

The authors would like to start by thanking the reviewers for taking the time to review our work and for their positive and helpful reviews. Below, we have addressed each of your comments, described the changes we have made in the manuscript, provided the line numbers of those changes within the tracked changes version, and provided the adjusted/added text in this document for your convenience. 

Reviewer #1 Comments:

Reviewer #1: Thank you for the opportunity to review this paper.

1. Introduction: Clear but could be shortened to be more concise. 

Response: Thank you for your appreciation of the clarity of this section. Within the context of PLOS ONE, which has a non-surgical specialty audience, it is more likely that readers will require all of the background provided and the complete rationale in order to appreciate the purpose of the work. Therefore, after carefully reviewing this section, we do not believe that it can be shortened without losing essential content. 

2. Methods: Clear. Power analysis performed. It is this reviewer's opinion that the one sample t-tests adds limited value. Instead, as noted by the authors, is it possible to rather evaluate accuracy in terms of clinical relevance?

Response: Thank you for your positive comment regarding our overall methods. Regarding the one sample t-test. We do agree with the reviewer that the most important consideration is how closely our method compares to existing methods that are reported in the literature and thus we have spent significant time in the discussion talking about this factor. However, we do believe that it is also important to compare our results to the intended target (i.e. the actual position and orientation we pre-operatively planned to achieve) as that is our technical goal and thus it is important to understand if our results statistically differ from that goal. We hope that the reviewer agrees with this explanation. We have added some of this justification into our methods section on lines 252-255, as seen below for convenience.

Changes to original text are underlined: “From a clinical feasibility perspective, it is most important to compare the resulting error values to error/accuracy values reported in the literature for previous methods as is done in the discussion section. However, from a technical development perspective, it is also important to compare the error values to the pre-operatively planned target placement. Therefore, for the PSG and final implant accuracy outcomes, single sample t-tests were conducted to determine if a statistically significant difference existed between the measured error values and a value of zero error (i.e. the pre-operative plan). Note that a significant difference only indicates the result statistically differs from the pre-operative plan but not that the error is clinically unacceptable.”

3. Results: Clear and concise. See comment with regards to the p-values in the methods section.

Response: Thank you for your positive comment. Regarding the p-values, as explained in response to the reviewer’s methodology comment, we do believe that conducting this single sample t-test is important in order to determine if our results are statistically different from our pre-operatively planned goal. 

4. Discussion: A major limitation of the study is the lack of a control group for comparison of the achieved accuracy between the minimal invasive and invasive approaches. This should be highlighted in the text. The discussion is somewhat long and can be shortened without loosing any relevance.

Response: Thank you for raising this concern. We did briefly mention this factor in our methodology: “No control specimens, treated using a traditional open surgical technique, were tested in this study as significant data are available in the literature [20,29–33,36,37] and thus the use of additional cadaveric remains could not be ethically justified.” However, we do appreciate that this is a limitation and should have been included in our limitations section. We have now added it to our limitations section on lines 326-332, as seen below for convenience. 

Changes to original text are underlined: “Third, a control group of specimens treated with a traditional open surgical technique was not included in this study as significant data are available in the literature detailing the accuracy of these procedures and thus it was not thought appropriate to use additional cadaveric specimens to obtain similar data. However, this decision did limit the ability to statistically compare the developed method to traditional approaches and to collect data on the accuracy of each surgical step, which are not available in the literature. Despite this, the data available from the literature was sufficient to evaluate the overall accuracy of the developed technique.”

Reviewer #2: Dear author,

thank you for your interesting paper about an innovative surgical technic of shoulder prosthesis implantation. However the application field in terms of indications seems limited (Young patients with mild OA) and not sure that the accuracy provided by the system is so acceptable than you concluded.

Response: Thank you for your comment regarding the innovative nature of the developed method. Regarding your first concern about the limited scope of the application, we would first like to clarify that it is focused on moderate not mild osteoarthritis (OA), which is a rapidly growing segment of the shoulder replacement surgical volume. To clarify, this method is not limited to younger patients – it can be used with any patient having moderate OA irrespective of age – the authors simply mentioned younger individuals because the less invasive nature of the technique will be particularly attractive for active young patients. We apologize for this confusion, there were a number of places in the manuscript where we accidentally conflated younger patients and patients with moderate OA, and although there is a correlation between these two factors, there are also older patients with moderate OA that this technique could be employed with. Therefore, we have made a number of small clarifications on lines 34-35 & 410 to make this more clear (included below for convenience).

Regarding the reviewer’s second comment about the acceptability of our achieved accuracy. We had originally hoped to achieve the same accuracy as patient specific guides used with an open procedure, thus achieving both significantly less invasiveness while maintaining accuracy as good as the current gold standard. However, although our accuracy did not match that of existing patient specific guides with open surgical technique, it did achieve results similar to what is still the most widely used technique, which is an open procedure without patient specific guides. Therefore, we do believe that our accuracy is acceptable as it is similar to what is widely used but with significantly less invasiveness. Additionally, as explained in our discussion, the tests were performed with specimens having mild OA because specimens with moderate OA could not be procured. Given that specimens with moderate OA would yield improved registration accuracy for our patient specific guides and this accuracy was found to be the most prominent factor in final implantation accuracy, we do expect to improve our accuracy when used on the intended patient cohort. With all of this said, we hope that it is clear that although the accuracy result is not as outstanding as we had initially hoped, it is still a significant achievement within the context of simultaneously minimizing surgical invasiveness. 

Changes to original text are underlined: “This novel technique could lead to a paradigm shift in shoulder arthroplasty for patient with moderate arthritis, which could significantly improve rehabilitation and functional outcomes.”

Changes to original text are underlined: “With these potential benefits and the current results in mind, this novel technique and technology has the potential to shift the paradigm in TSA for patients with moderate arthritis.”

I don't have any comments for the methodology that is quite rigorous, except for humeral rotation assessment that is unaccurate... or for other details...

Response: Thank you for your positive comment about the methodology. Regarding the humeral rotation, the authors would like to clarify that the method is not inaccurate in this rotation, rather, we simply chose not to have the technique control that rotation, as it would increase complexity for no benefit i.e. to control a rotation about which the implant is axisymmetric and thus its orientation is non-critical to the surgical outcome. Furthermore, traditional techniques also do not control this rotation, as it is left to the surgeon. Thus, we chose simply to replicate what is standard in terms of the degrees of freedom that are controlled. 

I have few remarks about the form:

- L30-31: SD values are missing

Response: To clarify, we have not omitted the SDs; instead, in order to include the relevant data for all rotations in the tight word limit of the abstract, we have included the range of errors across the various rotations plus or minus the range of standard deviations for those same rotations. Apologies for the confusion.

- L33: change "equal" for similar

Response: Thank you for pointing this out, we agree that equal is too precise. We have changed it with similar as you suggested.

- L81-86: these sentences could be removed because of the redundancy with the introduction especially

Response: Thank you for raising this concern. We have worked to shorten this section and removed redundancy but have not entirely removed it as it refers to a new surgical technique not discussed in the introduction and thus much of the included information is required. See lines 85-86 (included below for convenience).

Changes to original text are underlined: “However, not dislocating the joint when using this MI approach prevents the traditionally required en-face articular access; thus, a new paradigm and instrumentation are required for bone preparation and implantation.” 

- Fig 2: please flip the picture b, and change the arrangement of the 2 pieces of the PSG on the picture a, for easing the understanding of the operation of the PSG fixation (to get the same orientation of the visualization)

Response: Thank you for raising this point. We agree that a consistent orientation will help the reader follow the procedural process and so have made the suggested changes as well as making the two PSG components have distinct colours in each render to make it easier to associate each component in each figure pane. To further improve understanding of the process, we have added an additional piece of Supplemental Information in the form of a CAD rendered animation (titled S1 Animation.mp4) showing all of the steps from Patient Specific Guide insertion through to transhumeral tunnel drilling and guide removal. 

- Fig 3: same comment for the picture b

Response: Thank you for raising this point. We agree that a consistent orientation will help the reader follow the procedural process and so have made the suggested changes.

The nerve proximity at the level of the glenoid neck would have been interesting to assess too because the fixation of the PSG looks very prominent in this area...

Response: Thank you for suggesting this. The authors did consider measuring the relative position of the axillary nerve to the patient specific guide and instruments; however, we chose not to do this because our clinical PI and a previous surgical trainee had already conducted a study investigating the posterior surgical approach including the proximity of the nerve to the incision location and this was published in the Journal of Shoulder and Elbow Surgery (reference 34 in our manuscript). Therefore, we did not believe there was added value to retaking these measurements.

---

## [Decision Letter · Decision Letter 1]

5 May 2021

Development and ex-vivo assessment of a novel patient specific guide and instrumentation system for minimally invasive total shoulder arthroplasty

PONE-D-20-37057R1

Dear Dr. Giles,

We’re pleased to inform you that your manuscript has been judged scientifically suitable for publication and will be formally accepted for publication once it meets all outstanding technical requirements.

Kind regards,

Alejandro A. Espinoza Orías, PhD

Academic Editor

PLOS ONE

Additional Editor Comments (optional):

Reviewers' comments:

Reviewer's Responses to Questions

**Comments to the Author**

1. If the authors have adequately addressed your comments raised in a previous round of review and you feel that this manuscript is now acceptable for publication, you may indicate that here to bypass the “Comments to the Author” section, enter your conflict of interest statement in the “Confidential to Editor” section, and submit your "Accept" recommendation.

Reviewer #1: All comments have been addressed

Reviewer #2: All comments have been addressed

2. Is the manuscript technically sound, and do the data support the conclusions?

Reviewer #1: Yes

Reviewer #2: Yes

3. Has the statistical analysis been performed appropriately and rigorously? 

Reviewer #1: Yes

Reviewer #2: Yes

4. Have the authors made all data underlying the findings in their manuscript fully available?

Reviewer #1: Yes

Reviewer #2: Yes

5. Is the manuscript presented in an intelligible fashion and written in standard English?

Reviewer #1: Yes

Reviewer #2: Yes

6. Review Comments to the Author

Reviewer #1: Thank you dor submitting a revised version. You have addressed all my comments adequately. I have no further comments or suggestions.

Reviewer #2: (No Response)

7. PLOS authors have the option to publish the peer review history of their article (what does this mean?). If published, this will include your full peer review and any attached files.

Reviewer #1: No

Reviewer #2: No

---

## [Editor Report · Acceptance letter]

14 May 2021

PONE-D-20-37057R1 

Development and *ex-vivo* assessment of a novel patient specific guide and instrumentation system for minimally invasive total shoulder arthroplasty 

Dear Dr. Giles:

I'm pleased to inform you that your manuscript has been deemed suitable for publication in PLOS ONE. Congratulations! Your manuscript is now with our production department. 

Kind regards, 

on behalf of

Dr. Alejandro A. Espinoza Orías 

Academic Editor

PLOS ONE